# Asymptotic Analysis for the Effects of Anode Inlet Humidity on the Fastest Power Attenuation Single Cell in a Vehicle Fuel Cell Stack

**Yongfeng Liu** [1,2,*], **Jianhua Gao** [1,*], **Na Wang** [1] **and Shengzhuo Yao** [1]

1    Beijing Key Laboratory of Performance Guarantee on Urban Rail Transit Vehicles, School of Machine-Electricity and Vehicle Engineering, Beijing University of Civil Engineering and Architecture, Beijing 100044, China; 13811920259@163.com (N.W.); yaoshengzhuo@bucea.edu.cn (S.Y.)
2    State Key Laboratory of Automotive Safety and Energy, Tsinghua University, Beijing 100084, China
*    Correspondence: liuyongfeng@bucea.edu.cn (Y.L.); godistian@163.com (J.G.)

**Abstract:** A three-dimensional and isothermal anode relative humidity (ARH) model is presented and used to study the anode inlet humidity effects on the fastest power attenuation single cell in a vehicle fuel cell stack. The ARH model is based on the phenomenon that the anode is more sensitive than the cathode to water flooding. The pressure drop is considered in the ARH model, and saturation pressure is established by a pressure drop. Based on the pressure drop and relative humidity, simulations and tests are completed. First, the geometric model and computational grids are established, based on real structure of the proton exchange membrane fuel cell (PEMFC). Second, single cell distribution in the stack, test schematic and experimental conditions are demonstrated. Finally, polarization curves with 10 cells are displayed and discussed under these conditions that working temperature 70 °C, and diverse relative humidity (40%, 55%, 70%, 85%, and 100%). The test results of 34 cm$^2$ fuel cell stack are compared against simulation results. The results show that C10 (the single cell with the farthest distance from the gas inlet) power attenuation is the fastest and that its performance is the poorest under the experimental conditions. The polarization curves predicted by the ARH model indicate fairly good coherence with the experimental results, compared against the Fluent original model. The ARH model calculation deviation is 28% less than the Fluent model at 360 mA·cm$^{-2}$ for a relative humidity of 85%. The current density distribution is almost uniform, and membrane water content is negatively affected by high humidity.

**Keywords:** proton exchange membrane fuel cell; pressure drop; relative humidity; power attenuation

## 1. Introduction

In recent years, hydrogen as a renewable clean energy is widely used in hydrogen internal combustion engines and proton exchange membrane fuel cell (PEMFC) vehicles, due to the excessive consumption of fossil fuels and the emissions of pollutants [1–3]. PEMFC vehicles are considered to be a feasible alternative to internal combustion engines resulting from their high power density, high efficiency (85%) and zero pollution [4]. As a future development object of automotive power source, PEMFC has received extensive attention [5–8]. For the PEMFC stack, as the core component of fuel cell vehicles, its cost and life are the pivotal factors for fuel cell commercialization [9–12]. The excellent humidity of the fuel cell has a direct positive effect on single cell power attenuation and the stack performance, so as to prolong the cell life. However, on the one hand, it can lead to the serious drying, even cracking, of the proton exchange membrane, when the fuel cell humidification is insufficient [13]. On the other hand, water flooding phenomenon can be caused by excessive humidification, which

leads to local hot spots due to gas accumulation in the fuel cell, and shortening the fuel cell life. Thus, it is beneficial to set appropriate humidification of the gas to ensure good conductivity of the proton exchange membrane and to improve water management system of the fuel cell [14].

At present, the humidification methods applied in experiments include internal humidification [15], external humidification [16,17] and self-humidification [18,19]. For internal humidification, liquid water is transferred in the membrane according to the effect of concentration difference. This method is convenient to operate, but the problems of hysteresis and inaccuracy are obvious. For external humidification, one commonly used method is liquid water humidification, which is simple and efficient; however, this method has serious flooding problems, which leads to the deterioration of PEMFC performance. Another method is bubble humidification, the issue for which is controlling the temperature and humidity when the gas enters. In addition, self-humidification is affected by working conditions, and it is only useful for low-power applications. External humidification is superior to internal humidification and self-humidification, especially under high power and high temperature conditions [20].

To some extent, relative humidity as an inlet parameter can influence the water management, thus affecting the PEMFC performance. Many researchers have focused on the relative humidity effects on the PEMFC performance from experiment and calculation. As a customary research method, the experimental research is not only intuitive, but it also has a high reliability. Zhang et al. [21] designed an air humidifier for evaluating the humidification performance under different operation conditions. Pei et al. [22] revealed the effects of different anode and cathode gas humidity levels to determine the cause of the flooding. Dilek et al. [23] proposed that the water content, temperature, and conductivity of the proton exchange membrane could be improved by increasing the relative humidity and temperature of the cell. R Eckl et al. [24] revealed that gas humidification played a crucial role in the low power range for high operating temperatures. Also, they explained the physical and chemical mechanism applications by observing the experimental phenomenon of fuel cell in two kinds of extreme cases, dry and flooded.

Compared to experimental studies, simulations have a wide range of applications. They can greatly reduce material and financial resources, but also effectively shorten the test time required. Jeon et al. [25] evaluated the PEMFC using the system efficiency calculation method at different working temperatures and inlet relative humidities, and stated that the system efficiency was improved more availably by changing the relative humidity rather than the operating temperature. Aleksandra et al. [26] studied the effects of the cathode inlet temperature and relative humidity on performance, and found that the residual heat in the fuel cell could affect the fuel cell performance, and that lower temperature and higher relative humidity resulted in better fuel cell performance. The relative humidity influence on PEMFC vehicle operation has also caused Jeon et al. [27] great interests. They concluded that low humidity was beneficial in reducing costs and the overall weight of the humidifier in the external cell cathode, according to serious flooding phenomena.

As mentioned in the above research, the relative humidity significant effects on the PEMFC water management system and performance has been confirmed. However, with the fast development of PEMFC, the emerging stack combined by many single fuel cells satisfied the need of high power; meanwhile, homogeneity problems have arisen. This paper takes into account the pressure drop about the fastest power attenuation single cell in the stack, and indicates the saturation pressure by the pressure drop, and comprehensively analyzes the fuel cell performance from two aspects of calculation and experiment. Firstly, a three-dimensional anode relative humidity (ARH) model is put forth. Then, the geometric model and grid computing are established, and the ARH model is coupled into computational fluid dynamics software for calculation. Next, tests are conducted on the basis of an experimental platform under these conditions: a cell working temperature of 70 °C and an inlet pressure of 0.1 MPa with a relative humidity of 40%, 55%, 70%, 85%, and 100%. Finally, the experimental results of the Fluent model and the ARH model are compared. The counters of fuel cell species are displayed and analyzed.

## 2. Calculation

*2.1. ARH Model Description*

The mass conservation equation is applied to the diffusion of the gas mixture, which is listed as:

$$\frac{\partial}{\partial x}\left(\rho \vec{u}\right) = S_m \tag{1}$$

where $\rho$ represents the density of gas (kg m$^{-3}$) and $u$ is the velocity vector (m s$^{-1}$). $S_m$ is the sum of the hydrogen, oxygen and water source terms, as listed below:

$$S_m = S_{H_2} + S_{O_2} + S_{H_2O} \tag{2}$$

$$S_{H_2} = -\frac{M_{H_2}}{2F}R_a \tag{3}$$

$$S_{O_2} = -\frac{M_{O_2}}{4F}R_c \tag{4}$$

$$S_{H_2O} = \frac{M_{H_2O}}{2F}R_c \tag{5}$$

where $R_a$ represents the hydrogen exchange current densities (A m$^{-3}$), $R_c$ represents the oxygen exchange current densities (A m$^{-3}$). $M_{H_2}$ is the hydrogen molecular weight (kg mol$^{-1}$), $M_{O_2}$ is the oxygen molecular weight (kg mol$^{-1}$), and $M_{H_2O}$ is the water molecular weight (kg mol$^{-1}$). $F$ represents the Faraday constant.

The momentum conservation equation and the momentum source term are given by:

$$\frac{\partial(\varepsilon \rho \vec{u})}{\partial t} + \nabla \cdot (\varepsilon \rho \vec{u} \vec{u}) + \varepsilon \nabla P - \nabla \cdot (\varepsilon \mu \nabla \vec{u}) = S_u \tag{6}$$

$$S_u = -\frac{\mu}{K}\vec{u} \tag{7}$$

where $S_u$ is the momentum source term in the catalyst layer and the gas diffusion layer of PEMFC, and it is 0 in other domains. $\varepsilon$ represents the porosity coefficient. $P$ is the pressure (Pa), $T$ is the temperature (K), and $\mu$ indicates the mixture mean viscosity (kg (m s)$^{-1}$). $K$ is the permeability.

The diffusivity of gas phase species is expressed as:

$$D_e = \varepsilon^{1.5}D_g \tag{8}$$

Membrane conductivity and the osmotic drag coefficient are given by (9) and (10). Due to excellent stability and conductivity, Nafion membrane is used for PEMFC, working as a kind of perfluorinated sulfonic acid membranes. The diffusion flux is given by (11).

$$\sigma_m = (0.514\lambda - 0.326)e^{1268\left(\frac{1}{303} - \frac{1}{T}\right)} \tag{9}$$

$$n_d = 2.5\frac{\lambda}{22} \tag{10}$$

$$J_d = -\frac{\rho_m}{M_m}M_{H_2O}D(\lambda)\nabla\lambda \tag{11}$$

where $D_e$ and $D_g$ are diffusivity of the effective gas species and the gas species mass (m$^2$ s$^{-1}$), respectively. $\sigma_m$, $M_m$, and $\rho_m$ are the conductivity, equivalent weight (kg kmol$^{-1}$), and density (kg m$^{-3}$) of the membrane. $\lambda$ is the water content.

Different from the traditional internal combustion engine, the fuel cell stack replaces the combustion chamber, and its efficiency is not limited by the Kano cycle. In addition to the traditional internal combustion requirements of the air filter and pressure for the supply system, the fuel cell stack needs to be humidified [28,29]. In order to effectively prevent the water flooding phenomenon of the vehicle fuel cell, Pei et al. [30] carried out the relevant research. They found that the reverse osmosis of the cathode water produced made the anode less resistant to water flooding, so they focused on anode water flooding. Through a series of theoretical analysis and experimental verification, they concluded that the pressure drop had significant effects on water flooding and obtained pressure drop law affected by current and other parameters.

$$\Delta P_f = \frac{1.1748 \times 10^{-9}(C_w + C_d)Le^{T/275.7}T}{n(C_wC_d)^3(P_{H_2} - P_s)P_{H_2}^{0.0263}}(\lambda_{H_2} - 0.5)I \tag{12}$$

Membrane water diffusivity in ARH model is given by:

$$D(\lambda) = f(\lambda)e^{2416(\frac{1}{303} - \frac{1}{T})} \tag{13}$$

$$\lambda = 0.043 + 17.18a - 39.85a^2 + 36a^3 \ (a < 1)$$
$$\lambda = 14 + 1.4(a - 1) \ (a > 1) \tag{14}$$

$$a = \frac{P_w}{P_s} \tag{15}$$

The saturation pressure in the above Equation (15) could be obtained by Equation (12), not the empirical equation in Fluent.

$$P_s = P_{H_2} - \frac{1.1748 \times 10^{-9}(C_w + C_d)^2 Le^{T/275.7}T}{n(C_wC_d)^3 P_{H_2}^{0.0263}\Delta P_f}(\lambda_{H_2} - 0.5)I \tag{16}$$

$$RH = \frac{P_i}{P_{sat}}X_{H_2O} \tag{17}$$

where $C_w$, $C_d$, $L$, $n$, $P_{H_2}$, $P_s$, $\lambda_{H2}$, $I$, $RH$, $a$, $X_{H_2O}$, $P_e$, and $P_w$ represent the flow channel width (cm), flow channel depth (cm), flow channel length (cm), the number of flow channels, anode inlet pressure (kPa), the saturation pressure (kPa), anode gas excess coefficient, current (A), relative humidity (%), water activity, vapor molar fraction, the local pressure (kPa), and the water vapor pressure (kPa), respectively.

It is required that all imports are represented as a mass-flow-inlet and exports as a pressure-outlet for boundary conditions. The mass flow rate is:

$$M_A = \frac{\lfloor 18X_{H_2O} + 2(1 - X_{H_2O})\rfloor K_{H_2} \cdot i \cdot A \cdot \lambda_{H_2}}{2(1 - X_{H_2O})} \tag{18}$$

$$M_C = \frac{\lfloor 18X_{H_2O} + 29(1 - X_{H_2O})\rfloor K_{O_2} \cdot i \cdot A \cdot \lambda_{O_2}}{0.23 \times 29(1 - X_{H_2O})} \tag{19}$$

where $A$ represents the active area (m$^2$), $i$ represents the current density (A m$^{-2}$), $K_{H_2}$ is the hydrogen electrochemical equivalent (kg(A s)$^{-1}$) and $K_{O_2}$ is the oxygen electrochemical equivalent (kg(A s)$^{-1}$), $\lambda_{O2}$ is the cathode gas excess coefficient.

The pressure drop is considered in the calculation of relative humidity in the ARH model, so that the relative humidity is obtained by inlet pressure and saturation pressure. In the process of the calculation, the outlet pressure is adjusted in time. For unknown PEMFC, the coefficients in Equations (12) and (16) can be obtained by fitting the data. It can also be said that the anode pressure drop can

be ascertain by the experiment pressure drop data of certain fuel cell temperatures, inlet pressures, and humidity.

## 2.2. Simulation Schematic

The fuel cell numerical analysis schematic is shown in Figure 1. First, the fuel cell geometric model with a serpentine flow field is established, based on the actual experimental cell. The mesh generation is carried out, and its quality is checked. Next, the PEMFC model is defined in Fluent 15.0, and the ARH model is coupled into the PEMFC module to calculate; what is more, the fluid grids and boundary conditions are set. Then, solvers, including the iterative steps and convergence conditions, are set up. Finally, the pressure and other parameters are adjusted to satisfy the convergence conditions in time, and then the results are processed and unreasonable solutions are removed.

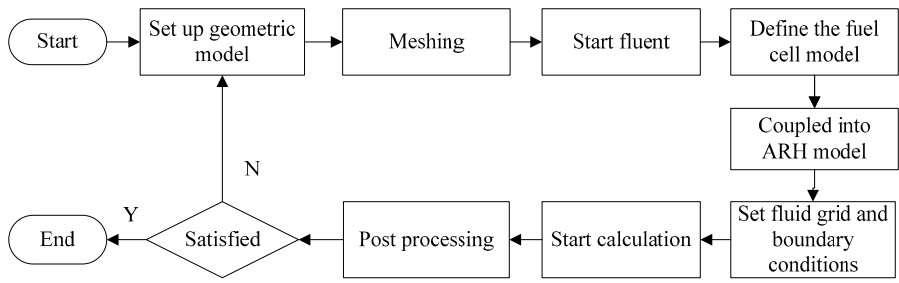

**Figure 1.** The fuel cell numerical analysis schematic.

## 2.3. Geometry and Mesh Generation

As shown in Figure 2a, the PEMFC geometry includes the proton exchange membrane, diffusion layers, catalyst layers, flow channels, and collector plates. The difference is that Falcao et al. [31] the geometry does not include collector plates, in order to shorten the simulation time. The entire mesh generation is shown in Figure 2b. The anode and cathode flow channel mesh generation is respectively listed in Figure 2c,d for better observation. Non-conformal mesh is used at the interface to reduce the workload, and to improve mesh quality. The number of elements is almost up to 1,600,000. The number of anode and cathode flow channels elements is 159,653 and 138,246, respectively. The body mesh generation is non-uniform as Rahimi-Esbo [32] presented, because of the dimensional difference. Collector plates and channels have the same elements Tet/Hybrid and type TGrid, and other parts use the same elements Hex and type Map. It is assumed that the gas mixture is ideal and incompressible, and that the reactions are homogeneous, resulting from low flow rate and low pressure. The model is isothermal and single-phase. The parameters of geometrical configurations, transport properties, and electrochemical properties in the ARH model are listed in Table 1.

**Table 1.** Parameters in the anode relative humidity (ARH) model.

| Parameters | Value | Unit |
|---|---|---|
| Active area | 0.0034 | $m^2$ |
| Thickness of the membrane | 0.00005 | m |
| Thickness of the diffusion layer | 0.0002 | m |
| Thickness of the catalyst layer | 0.00001 | m |
| Length of the flow channel | 0.05 | m |
| Width of the flow channel | 0.0012 | m |
| Depth of the hydrogen channel | 0.0006 | m |
| Depth of the oxygen channel | 0.0008 | m |
| Thickness of the collector plate | 0.002 | m |
| Number of serpentine turns | 5 | |
| Length of the single channel turn | 0.05 | m |
| Thermal conductivity of the membrane | 0.4 | $W \, (m \, K)^{-1}$ |
| Thermal conductivity of the gas diffusion layer | 1.2 | $W \, (m \, K)^{-1}$ |

**Table 1.** *Cont.*

| Parameters | Value | Unit |
|---|---|---|
| Thermal conductivity of the catalyst layer | 1.5 | W (m K)$^{-1}$ |
| Thermal conductivity of the current collector | 20 | W (m K)$^{-1}$ |
| Electrical conductivity of the gas diffusion layer | 2500 | (ohm m)$^{-1}$ |
| Electrical conductivity of the catalyst layer | 2500 | (ohm m)$^{-1}$ |
| Electrical conductivity of the current collector | 20,000 | (ohm m)$^{-1}$ |
| Porosity of the gas diffusion layer | 0.5 | |
| Porosity of the catalyst layer | 0.28 | |
| Membrane equivalent weight | 1100 | kg kmol$^{-1}$ |
| Hydrogen reference exchange current density | 4000 | A m$^{-2}$ |
| Anode reference concentration | 1 | kmol m$^{-3}$ |
| Anode transfer coefficient | 0.5 | |
| Oxygen reference exchange current density | 5.75 | A m$^{-2}$ |
| Cathode reference concentration | 1 | kmol m$^{-3}$ |
| Cathode transfer coefficient | 0.5 | |
| Open circuit voltage | 0.95 | V |
| Leakage current | 0 | A |
| Electrochemical equivalent of hydrogen | $1.05 \times 10^{-8}$ | kg (A s)$^{-1}$ |
| Electrochemical equivalent of oxygen | $8.29 \times 10^{-8}$ | kg (A s)$^{-1}$ |
| Reference diffusivity of hydrogen | $9.15 \times 10^{-5}$ | m$^2$ s$^{-1}$ |
| Reference diffusivity of oxygen | $2.2 \times 10^{-5}$ | m$^2$ s$^{-1}$ |
| Reference diffusivity of water | $2.56 \times 10^{-5}$ | m$^2$ s$^{-1}$ |
| Anode catalyst layer surface/volume ratio | $2 \times 10^6$ | m$^{-1}$ |
| Cathode catalyst layer surface/volume ratio | $1 \times 10^7$ | m$^{-1}$ |
| Operating temperature | 70 | °C |
| Anode relative humidity | 40, 55, 70, 85, 100 | % |
| Cathode relative humidity | 100 | % |

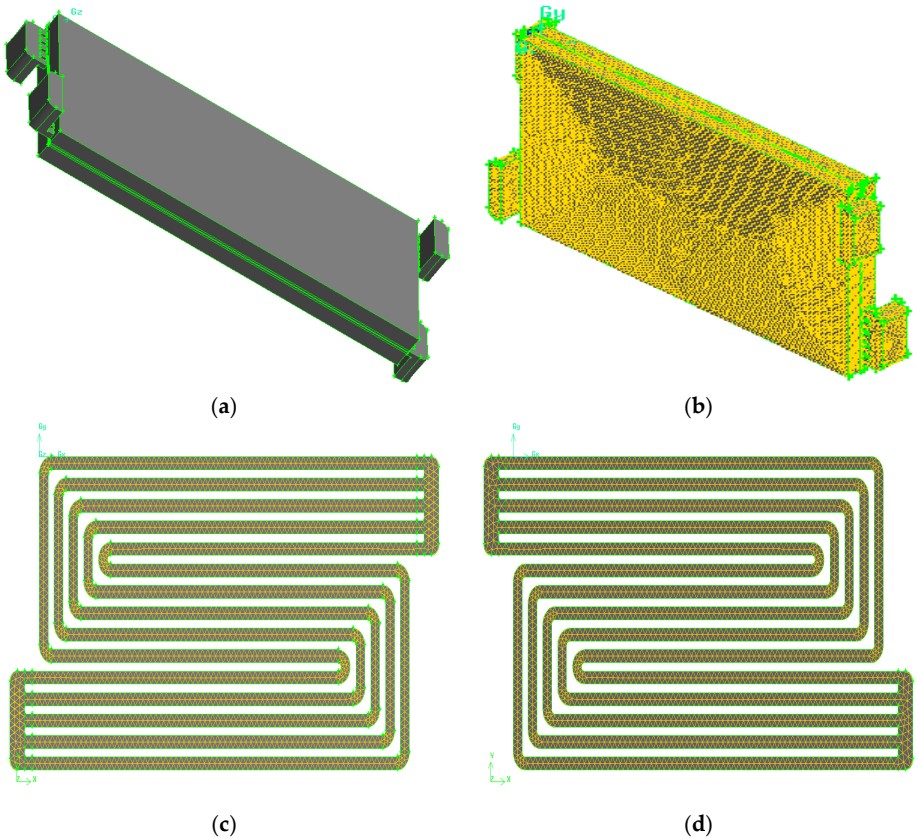

(a)

(b)

(c)

(d)

**Figure 2.** The proton exchange membrane fuel cell (PEMFC) geometry and mesh. (**a**) PEMFC geometry;
(**b**) PEMFC mesh generation; (**c**) The anode flow channels; (**d**) The cathode flow channels mesh.

For the experimental fuel cell stack, if the structure and channels location are unknown, the ARH model parameters can be estimated by performing the twice test in different parameters. The effective area of MEA could also be estimated by this method.

## 3. Experimental Set-Up

### 3.1. Apparatus and Schematic

Figure 3 shows the fuel cell stack and the schematic of single cell distribution. Figure 3a is the 10 single-cell stack and the test platform. The stack, composed of 10 single cells, uses hydrogen as a fuel at the anode, and oxygen as the oxidant at the cathode, and the heater band is used to maintain the temperature. Figure 3b shows the position of the inlet and outlet. The experiment and simulation focused on the fastest power attenuation single cell in the vehicle fuel cell stack.

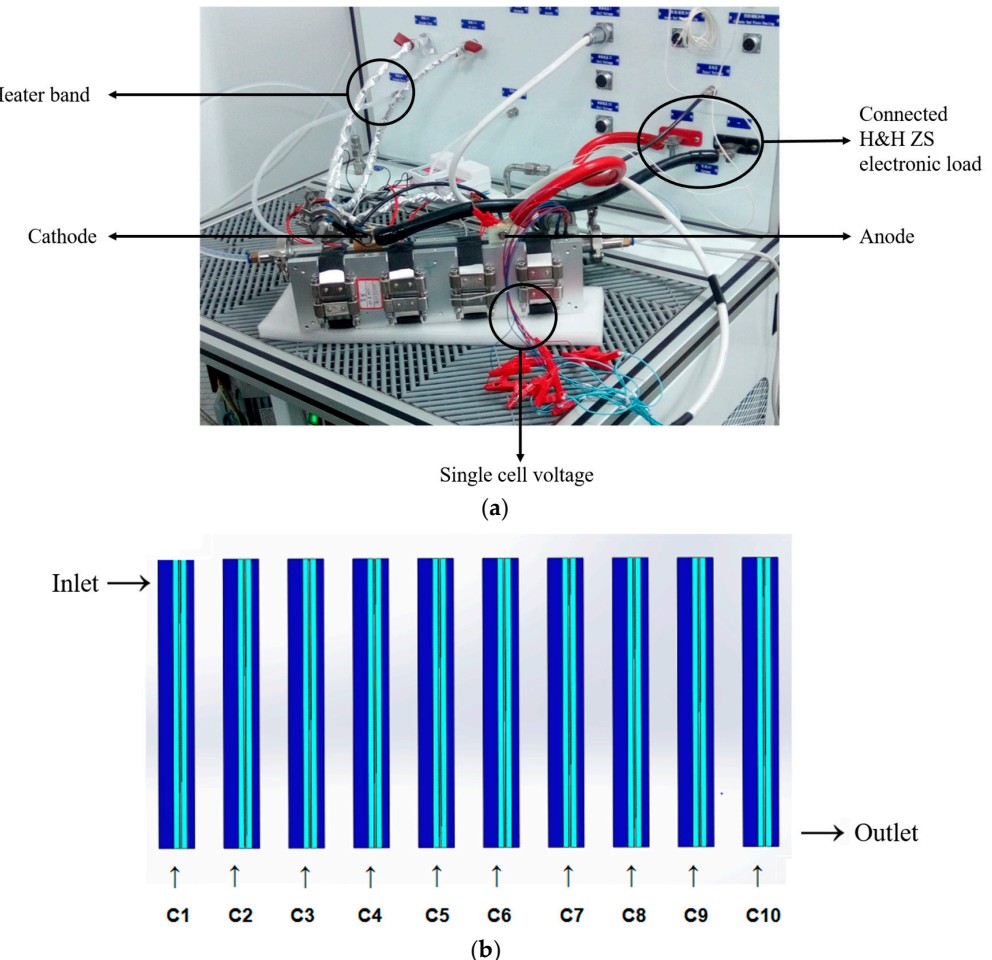

**Figure 3.** (**a**) Fuel cell stack and test platform; (**b**) Single cell distribution.

The fuel cell test system was made up of test elements, as shown in the test schematic in Figure 4. The test system power was up to 10 kW. There were two exit load systems in this experiment. On the one hand, the H&H ZS electronic load was used for providing load current. It takes advantage of the programming controls, and its control models are constant current, constant voltage, and constant power. On the other hand, RXN-3010D was utilized as a constant-current source in the measurement of membrane electrode parameters. The rotameter, pressure sensor, humidification sensor, and temperature sensor were used to detect the flow rate, pressure, humidification, and the temperature of the gas, respectively. Gas was humidified by a bubble humidifier. The fuel cell system was cooled by water cycle cooling. The pressure was controlled by the back-pressure valve at the

outlet. In order to prevent flooding and to ensure the inlet temperature, water-gas separators and a heater band were installed at the inlet. Normally, the temperature of the heater band is usually about 5 °C above the inlet temperature, to prevent gas cooling and liquefaction. Moreover, in order to ensure experimental safety, the alarm valve turns on and test bench is stopped immediately once hydrogen leakage exceeds the alarm threshold.

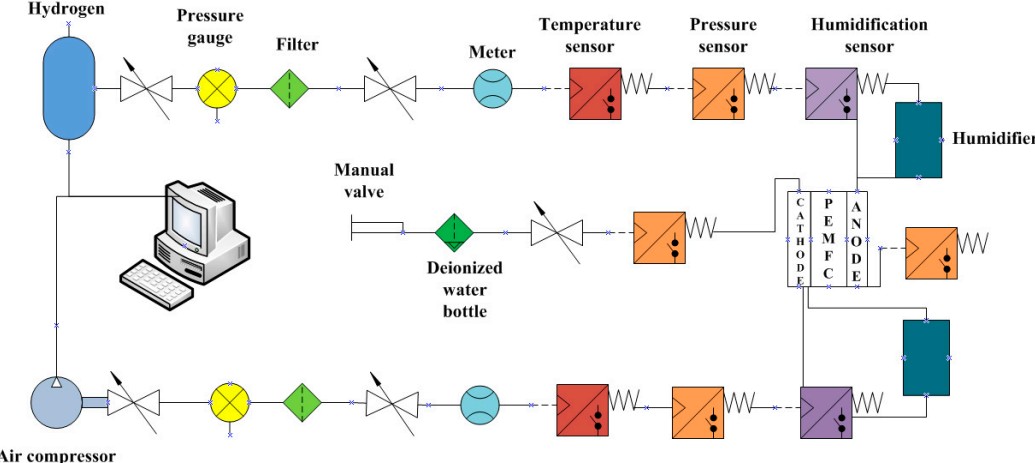

**Figure 4.** Test schematic. Relative humidity and other parameters are controlled and recorded by a personal computer (PC) and instances of pressure drop are taken manually.

### 3.2. Experimental Conditions

The ambient temperature of this experiment was 25 °C, and the ambient air relative humidity was 30%. Each parameter was controlled separately and they did not affect each other. Pure hydrogen and air were supplied and humidified by an external humidifier, and their flow rates were controlled by the gas flow meters. The excess coefficients of hydrogen and oxygen were respectively 1.4 and 2. The initial flow rates of hydrogen and oxygen were 2 L/min and 5 L/min. The experiment was performed under these conditions in that the operation temperature was 70 °C, the inlet pressure was 0.1 MPa, and the relative humidity was 40%, 55%, 70%, 85%, and 100% (with respective gas inlet temperature of 50 °C, 57 °C, 62 °C, 66 °C, and 70 °C). Polarization curves could then be drawn at the same operation temperature and at various gas inlet temperatures.

## 4. Results and Discussion

### 4.1. Polarization Curves with 10 Single Cells

Figure 5 presents the polarization curves with 10 cells for an operating temperature of 70 °C, a gas inlet pressure of 0.1 MPa and an elevated anode relative humidity of 40%, 55%, 70%, 85%, and 100%. Tables 2–6 show the dynamic performance at different relative humidity about the voltage drop. It can be found that the voltage was in the range of 0.551 V~0.973 V, while the current density was 0~360 mA/cm². Moreover, the maximum voltage appeared on single cells in different locations in the stack, because of different current densities. Around 0 current density, the maximum voltage appeared at C4 or C5, which was in the middle of stack. Oppositely, for a maximum current density of 360 mA/cm², the maximum voltage almost focused on a single cell of the gas inlet inside, due to a lower flow rate and the accumulation of gas. Jang et al. [33] also came to similar conclusions. They revealed that the cell performance on both sides of the stack was slightly better than that of the center cell. Additionally, the voltage differences between the single cells increased as the current density increased, because this was related to the increase of the reaction rate. The water flooding phenomenon did not occur because the experimental maximum current density almost did not reach the concentration polarization area.

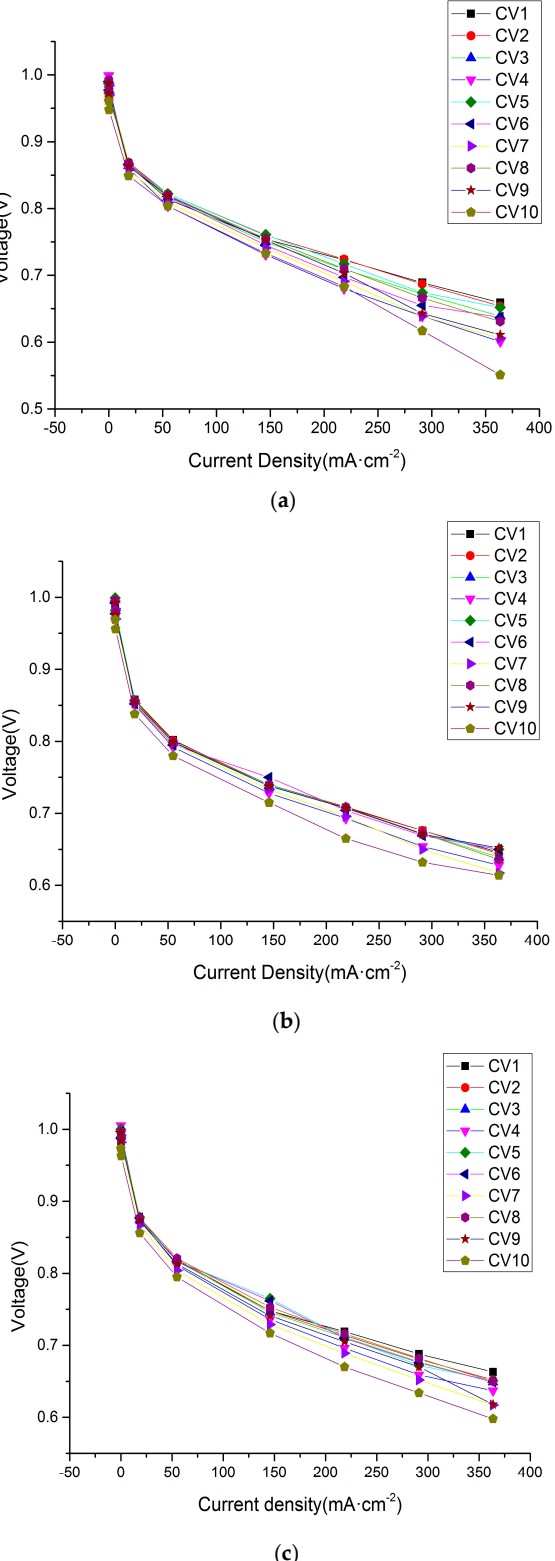

**Figure 5.** *Cont.*

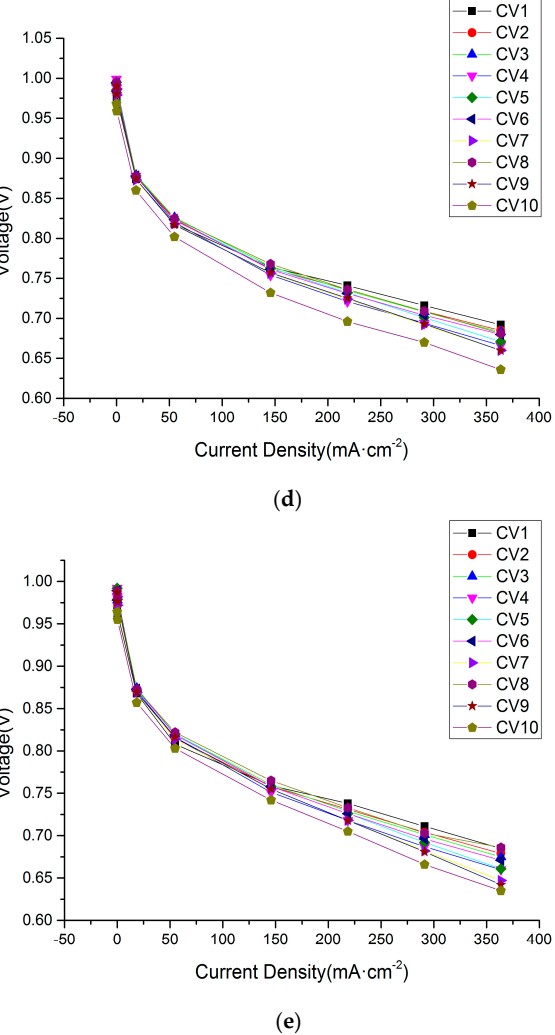

(d)

(e)

**Figure 5.** Polarization curves with 10 single cells with different relative humidity. The inlet pressure is 0.1 MPa. Working and ambient temperatures are 70 °C and 25 °C, respectively. (**a**) RH: 40%; (**b**) RH: 55%; (**c**) RH: 70%; (**d**) RH: 85%; (**e**) RH: 100%.

**Table 2.** Dynamic performance at different relative humidities (40%).

| Current Density (mA/cm$^2$) | CV1 | CV2 | CV3 | CV4 | CV5 | CV6 | CV7 | CV8 | CV9 | CV10 | Voltage |
|---|---|---|---|---|---|---|---|---|---|---|---|
| 0.012 | 0.978 | 0.982 | 0.992 | 0.999 | 0.987 | 0.990 | 0.984 | 0.992 | 0.987 | 0.961 | 9.840 |
| 18.494 | 0.865 | 0.869 | 0.864 | 0.860 | 0.867 | 0.866 | 0.858 | 0.867 | 0.863 | 0.849 | 8.613 |
| 54.861 | 0.814 | 0.822 | 0.817 | 0.804 | 0.822 | 0.818 | 0.809 | 0.820 | 0.819 | 0.804 | 8.109 |
| 145.783 | 0.754 | 0.760 | 0.750 | 0.731 | 0.761 | 0.745 | 0.741 | 0.755 | 0.753 | 0.733 | 7.459 |
| 218.502 | 0.723 | 0.724 | 0.709 | 0.680 | 0.717 | 0.697 | 0.691 | 0.710 | 0.703 | 0.683 | 7.010 |
| 291.222 | 0.689 | 0.687 | 0.672 | 0.639 | 0.674 | 0.655 | 0.639 | 0.666 | 0.643 | 0.617 | 6.561 |
| 363.595 | 0.659 | 0.654 | 0.639 | 0.601 | 0.652 | 0.637 | 0.606 | 0.631 | 0.611 | 0.551 | 6.158 |
| Voltage drop (V) | 0.319 | 0.328 | 0.353 | 0.398 | 0.335 | 0.353 | 0.378 | 0.361 | 0.376 | 0.410 | 3.683 |

As shown in Figure 5a and Table 2, the maximum voltage difference was 0.108 V between C1 and C10 at a current density of 360 mA/cm$^2$. The maximum voltage differences generated in Figure 5b–e were between two cells, which were C1 and C10, the same as Figure 5a, which respectively were 0.044 V, 0.065 V, 0.056 V, and 0.05 V at a current density of 360 mA/cm$^2$. It was obvious that the voltage differences between the cells gradually decreased as the relative humidity increased, resulting from the increase of ionic conductivity and resistance reduction [26]. It also could be observed that C10 voltage was the lowest, and its performance was the poorest for different current densities and relative

humidity. This may be because the amount of gas reaching C10 was relatively small, and the reaction rate was low, due to the pressure drop.

**Table 3.** Dynamic performance at different relative humidities (55%).

| Current Density (mA/cm$^2$) | CV1 | CV2 | CV3 | CV4 | CV5 | CV6 | CV7 | CV8 | CV9 | CV10 | Voltage |
|---|---|---|---|---|---|---|---|---|---|---|---|
| 0.012 | 0.988 | 0.993 | 0.997 | 0.992 | 0.999 | 0.992 | 0.984 | 0.996 | 0.992 | 0.969 | 9.891 |
| 18.494 | 0.858 | 0.857 | 0.856 | 0.851 | 0.857 | 0.852 | 0.853 | 0.856 | 0.854 | 0.838 | 8.513 |
| 54.869 | 0.802 | 0.801 | 0.798 | 0.792 | 0.798 | 0.795 | 0.797 | 0.799 | 0.799 | 0.780 | 7.922 |
| 145.790 | 0.738 | 0.740 | 0.738 | 0.728 | 0.739 | 0.750 | 0.733 | 0.740 | 0.738 | 0.715 | 7.326 |
| 218.502 | 0.708 | 0.709 | 0.708 | .693 | 0.708 | 0.704 | 0.696 | 0.708 | 0.707 | 0.665 | 6.973 |
| 291.222 | 0.676 | 0.676 | 0.672 | 0.654 | 0.671 | 0.669 | 0.650 | 0.672 | 0.671 | 0.632 | 6.614 |
| 363.595 | 0.646 | 0.644 | 0.639 | 0.628 | 0.650 | 0.650 | 0.617 | 0.636 | 0.652 | 0.614 | 6.264 |
| Voltage drop (V) | 0.342 | 0.349 | 0.358 | 0.364 | 0.349 | 0.342 | 0.367 | 0.360 | 0.340 | 0.355 | 3.628 |

**Table 4.** Dynamic performance at different relative humidity (70%).

| Current Density (mA/cm$^2$) | CV1 | CV2 | CV3 | CV4 | CV5 | CV6 | CV7 | CV8 | CV9 | CV10 | Voltage |
|---|---|---|---|---|---|---|---|---|---|---|---|
| 0.004 | 0.992 | 0.999 | 1.001 | 1.005 | 1.000 | 0.997 | 0.992 | 0.997 | 0.995 | 0.973 | 9.941 |
| 18.494 | 0.874 | 0.875 | 0.874 | 0.874 | 0.878 | 0.877 | 0.867 | 0.877 | 0.873 | 0.856 | 8.712 |
| 54.861 | 0.817 | 0.819 | 0.818 | 0.811 | 0.817 | 0.817 | 0.804 | 0.821 | 0.813 | 0.795 | 8.105 |
| 145.783 | 0.748 | 0.748 | 0.746 | 0.736 | 0.765 | 0.762 | 0.729 | 0.753 | 0.741 | 0.717 | 7.393 |
| 218.495 | 0.719 | 0.714 | 0.711 | 0.696 | 0.711 | 0.710 | 0.689 | 0.716 | 0.705 | 0.670 | 6.985 |
| 291.222 | 0.688 | 0.681 | 0.677 | 0.659 | 0.672 | 0.676 | 0.652 | 0.682 | 0.670 | 0.634 | 6.649 |
| 363.595 | 0.663 | 0.652 | 0.649 | 0.637 | 0.651 | 0.649 | 0.617 | 0.650 | 0.618 | 0.598 | 6.335 |
| Voltage drop (V) | 0.329 | 0.347 | 0.352 | 0.368 | 0.349 | 0.348 | 0.375 | 0.347 | 0.377 | 0.375 | 3.606 |

**Table 5.** Dynamic performance at different relative humidities (85%).

| Current Density (mA/cm$^2$) | CV1 | CV2 | CV3 | CV4 | CV5 | CV6 | CV7 | CV8 | CV9 | CV10 | Voltage |
|---|---|---|---|---|---|---|---|---|---|---|---|
| 0.004 | 0.983 | 0.992 | 0.995 | 0.999 | 0.995 | 0.994 | 0.987 | 0.995 | 0.990 | 0.968 | 9.888 |
| 18.494 | 0.874 | 0.877 | 0.879 | 0.878 | 0.878 | 0.878 | 0.873 | 0.878 | 0.874 | 0.860 | 8.736 |
| 54.861 | 0.818 | 0.823 | 0.826 | 0.820 | 0.824 | 0.824 | 0.818 | 0.825 | 0.817 | 0.802 | 8.175 |
| 145.783 | 0.764 | 0.762 | 0.765 | 0.754 | 0.763 | 0.761 | 0.756 | 0.768 | 0.757 | 0.732 | 7.564 |
| 218.495 | 0.741 | 0.736 | 0.735 | 0.721 | 0.732 | 0.731 | 0.724 | 0.736 | 0.725 | 0.696 | 7.241 |
| 291.222 | 0.716 | 0.709 | 0.708 | 0.694 | 0.701 | 0.704 | 0.692 | 0.709 | 0.693 | 0.670 | 6.946 |
| 363.588 | 0.692 | 0.685 | 0.683 | 0.666 | 0.671 | 0.680 | 0.660 | 0.681 | 0.660 | 0.636 | 6.677 |
| Voltage drop (V) | 0.291 | 0.307 | 0.312 | 0.333 | 0.324 | 0.314 | 0.327 | 0.314 | 0.330 | 0.332 | 3.211 |

**Table 6.** Dynamic performance at different relative humidities (100%).

| Current Density (mA/cm$^2$) | CV1 | CV2 | CV3 | CV4 | CV5 | CV6 | CV7 | CV8 | CV9 | CV10 | Voltage |
|---|---|---|---|---|---|---|---|---|---|---|---|
| 0.004 | 0.972 | 0.983 | 0.988 | 0.992 | 0.992 | 0.989 | 0.982 | 0.989 | 0.987 | 0.964 | 9.828 |
| 18.494 | 0.868 | 0.871 | 0.874 | 0.871 | 0.872 | 0.873 | 0.869 | 0.873 | 0.869 | 0.857 | 8.687 |
| 54.861 | 0.808 | 0.815 | 0.820 | 0.816 | 0.820 | 0.819 | 0.814 | 0.822 | 0.816 | 0.803 | 8.139 |
| 145.783 | 0.759 | 0.758 | 0.760 | 0.751 | 0.758 | 0.758 | 0.755 | 0.765 | 0.755 | 0.742 | 7.546 |
| 218.502 | 0.738 | 0.731 | 0.729 | 0.718 | 0.726 | 0.726 | 0.718 | 0.733 | 0.718 | 0.705 | 7.187 |
| 291.222 | 0.711 | 0.704 | 0.701 | 0.687 | 0.692 | 0.696 | 0.682 | 0.703 | 0.681 | 0.666 | 6.876 |
| 363.595 | 0.685 | 0.679 | 0.675 | 0.660 | 0.661 | 0.671 | 0.647 | 0.686 | 0.642 | 0.635 | 6.577 |
| Voltage drop (V) | 0.287 | 0.304 | 0.313 | 0.322 | 0.331 | 0.318 | 0.335 | 0.303 | 0.345 | 0.329 | 3.251 |

*4.2. The Fastest Power Attenuation Single Cell*

Based on the above series of experimental data, an important conclusion is that C10 is a single cell with the poorest performance in the stack being obtained. The ARH model was applied to the C10 calculation to compare among the Fluent model results and experiment results, and to prove its reliability. In order to more clearly express the cell performance, polarization curves, and power density curves about the ARH model, the Fluent model and experiment are shown in Figure 6 under

these conditions of working temperature, 70 °C, and different anode relative humidities. It could be found that the voltage and power density changing tendencies had a good agreement. However, there existed obvious voltage differences and power density differences between the experiment data and the simulation results for different relative humidities.

The C10 polarization curves and power density curves are shown in Figure 6a under a relative humidity of 40%. It can be observed that the voltage was in the range of 0.5~1.0 V when the current density varied from 0 to 400 mA·cm$^{-2}$. There was a maximum voltage for the experiment data and the simulation model results when the current density was around 0, because the electrochemical reaction rate was low, due to a lower gas inlet flow rate. The voltage differences between the experimental data and the simulation model data increased first, and then they gradually decreased as the current density increased. The maximum voltage difference appeared at around the current density of 50 mA·cm$^{-2}$. This is because the activation loss enhancement is related to the proton transfer rate and voltage reduction. The current density and reaction rate improvement causes an activation loss enhancement [17]. More specifically, the ARH model results were more reliable because they were always closer to the experimental results than the Fluent model results. The error between the ARH model (0.534 V) and the experiment data (0.499 V) was almost 7% at around a current density of 400 mA·cm$^{-2}$. This is because the ARH model not only takes into account the pressure drop between the flow channels of the single cells, but it also considers the pressure drop among the single cells in the stack. Also, water saturation improved with the increase of current density, resulting in the adsorption rate and throughput rate of the H$^+$ rise on the proton exchange membrane. Compared with the research of Li [34], we concluded that there are almost no concentration regions from the presented power density curves, so that there is no flooding phenomenon.

In Figure 6b, the polarization curves and power density curves for a relative humidity of 55% are shown. Compared against the polarization curves in Figure 6a, the minimum voltage was from 0.499 V up to 0.599 V. This was relevant to the higher ionic conductivity and lower resistance, and this could be attributed to the higher proton exchange membrane water content and the relative humidity [35]. The Fluent model results, ARH model results, and experiment results had a similar tendency to those in Figure 6a, and the curves changed more gently as the current density increased. When the current density was 0, the Fluent model, ARH model, and experiment voltages respectively were 0.972 V, 0.97 V, and 0.969 V and there was no difference, as the gas had not yet begun to react. It can be seen that there was a sharp drop in voltage when the current density was in the range of 0~50 mA·cm$^{-2}$. During this stage, the proton exchange membrane water content depended largely on the gas humidification, due to the lower electrochemical reaction. Similarly, the maximum differences of voltage and power density occurred near the current density of 50 mA·cm$^{-2}$, and the differences of the voltage and power density between the Fluent model (0.873 V, 43.65 mW·cm$^{-2}$) and the experiment (0.78 V, 39 mW·cm$^{-2}$) were 0.093 V and 4.65 mW·cm$^{-2}$, which were bigger than those (0.047 V, 2.35 mW·cm$^{-2}$) between the ARH model (0.827 V, 41.35 mW·cm$^{-2}$) and the experiment. It could easily be observed that the ARH model values were closer to the experiment values, to better predict the cell performance. There was a softer drop in voltage when the current density was in the range of 50~400 mA·cm$^{-2}$. The water content mainly came from the electrochemical reaction, and it was almost not affected by gas humidification with the increase of the current density. The calculation precision improved for the ARH model, because it considers the saturation pressure based on the pressure drop of the real PEMFC structure, and not the conventional equation.

A proper relative humidity is responsible for improved membrane water content and ion conductivity [36]. As shown in Figure 6c, several representative data for a relative humidity 70% are expressed in the polarization curves and power density curves. It could be found that simulation data and experiment data matched well. The voltage and power density differences gradually decreased when the current density was bigger than 50 mA·cm$^{-2}$. Especially, the Fluent model data (0.643 V) was the closest to the experiment data (0.598 V) at the current density of 360 mA·cm$^{-2}$, and even the voltage error was reduced to 8%, the same as Takalloo [37] explained. As we expected, the ARH model

data (0.619 V) was closer to experiment data compared to the Fluent model data at 360 mA·cm$^{-2}$. The water contents of the proton exchange membrane were produced by the electrochemical reaction and gas humidification. Additionally, it was obvious that the voltage and power density differences were smaller than that in Figure 6a,b. This is because the current density and relative humidity improvement caused the saturation of proton exchange membrane improvement, thus facilitating ion conductivity and reducing the resistance.

As shown in Figure 6d, the polarization curves and power density curves under the relative humidity 85% are displayed. The simulation data and experiment data showed no significant difference at around the current density of 0. The voltage differences between the simulation data and experiment data gradually increased from 0 to 50 mA·cm$^{-2}$, due to the increase of gas humidification and pressure drop. During the low current density region, gas humidification plays a vital role in the water content of the proton exchange membrane. The simulation data were consistent with the experiment data, and their differences were gradually ignored with the increase of the current density (50~400 mA·cm$^{-2}$). It can be observed that ARH model result (0.645 V) was closer to experiment data (0.636 V) at around the current density of 360 mA·cm$^{-2}$, and even the relative accuracy was increased by 72% compared with the Fluent model result (0.669 V). Since PEMFC operation temperature was relatively high, the ionic conductivity influenced by the high relative humidity was considered. In addition to the operating temperature, saturation pressure in ARH model takes into account the pressure drop, hydrogen inlet pressure, and the structural parameters of the cell. Thus, the ARH model results are more reliable than the Fluent model. Jeon et al. [19] also confirmed that cell system efficiency improvement depends on the relative humidity rather than the operation temperature. Additionally, the fuel cell is likely to be prone to flooding, due to the high operation temperature and the relative humidity [38].

There are, respectively, three polarization curves and power density curves with a relative humidity of 100% in Figure 6e. The simulation results and experiment results in Figure 6e were similar to the Figure 6a–d results. There existed a sharp drop in voltage at a current density of 0~50 mA·cm$^{-2}$. As a result of a low electrochemical reaction, gas humidification was necessary to ensure water content on the membrane. When the current density was about 360 mA·cm$^{-2}$, the Fluent model voltage (0.692 V) increases by 9% more than the experimental voltage (0.635 V), and the ARH model voltage (0.649 V) was closer to the experiment value, and it almost increased by 3% compared to the experiment voltage. The voltage differences decreased as predicted in Figure 6a–c. However, the voltage differences in Figure 6e rose again compared to that in Figure 6d. This is because the proton exchange membrane was likely to be flooded under real experiment due to higher relative humidity, but simulation conditions were affected by pressure drop, whose tendencies were the same as Figure 6a–d. In addition, Machado et al. [39] found that relative humidity had a stronger influence on cell performance, especially at a lower voltage range, as the gradient of water activity controlled the adsorption strength.

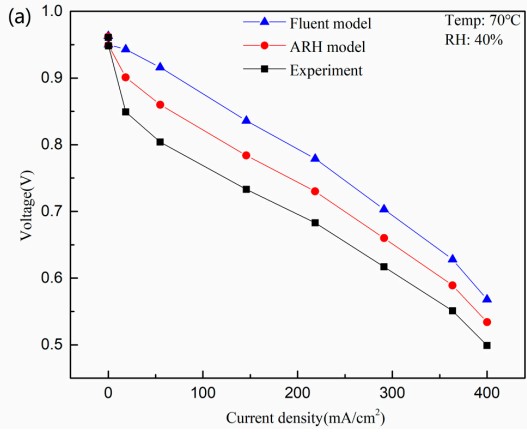

**Figure 6.** *Cont.*

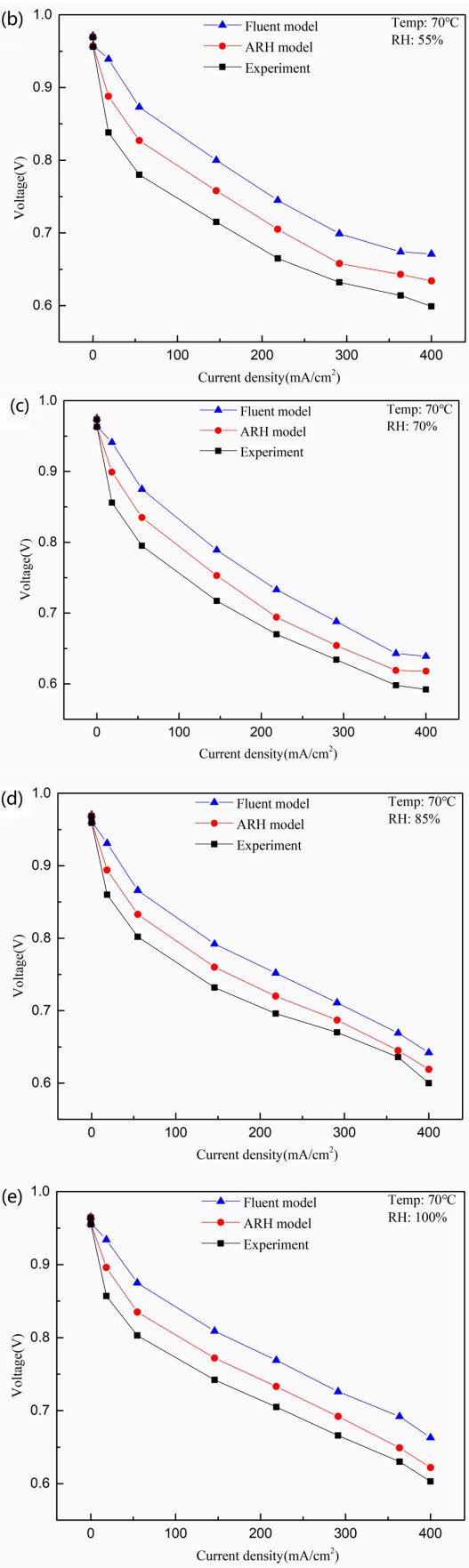

**Figure 6.** Comparison of the calculated model with the experiment results at different relative humidity. (**a**) RH: 40%; (**b**) RH: 55%; (**c**) RH: 70%; (**d**) RH: 85%; (**e**) RH: 100%.

### 4.3. Contours of Fuel Cell Species Distribution

The contours of the fuel cell species distribution including the current density, membrane water content, and molar concentrations of hydrogen and oxygen, are represented under the conditions of relative humidity 85% and 100% in Figure 7. It could be observed that the ARH model voltage value at a higher relative humidity was closer to real experiment value from the polarization curves, so that the contours of 85% and 100% relative humidity were chosen, and the operative voltage was 0.6 V.

The current density distribution on the proton exchange membrane surface is shown in Figure 7a,b. The current density distribution was uniform and similar to Rahimi-Esbo [32] research. Also, the pressure drop uniformity influenced current density distribution to some extent especially at the flow channel terminals. It could be found that the current density was in the range of $5.28 \times 10^2 \sim 8.95 \times 10^3$ at a relative humidity of 85%. The current density was in the range of $3.48 \times 10^2 \sim 8.93 \times 10^3$ at a relative humidity of 100%. It is obvious that there are bigger differences during the low current density region, and that the differences gradually narrow with the increase of current density. Figure 7c,d shows the counters of membrane water content distribution. It can be observed that membrane water content at a relative humidity of 85% was less than that at a relative humidity of 100% and that their results were opposite to the current density distribution. Additionally, the counters of molar concentrations of hydrogen and oxygen were respectively listed in Figure 7e–h. The variation range of hydrogen molar concentration was not large, and its tendency in anode channels was different, due to different relative humidities. From the gas inlet to the outlet in the anode channel, the molar concentration of hydrogen gas gradually decreased and the maximum and minimum concentrations were $4.31 \times 10^{-2}$ and $4.22 \times 10^{-2}$ at a relative humidity of 85%. A similar tendency has also emerged at a relative humidity of 100%, where the maximum molar concentration of hydrogen gas was $4.21 \times 10^{-2}$, and the minimum was $4.15 \times 10^{-2}$. This may be because the hydrogen flow rate and the mass fraction are relatively low, so the variation range is small [40]. The pressure drop and water content generated by a higher humidity have significantly effects on the molar concentration distribution, so the molar concentration values are smaller at relative humidities of 100%. Similarly, the oxygen molar concentrations are in the range of $7.47 \times 10^{-3} \sim 8.97 \times 10^{-3}$ and $7.06 \times 10^{-3} \sim 8.64 \times 10^{-3}$ at cathode channels for relative humidities of 85% and 100%, and their variation tendencies show a good agreement. Purushothama Chippar et al. [41] pointed out that simulation values such as current density and molar concentration at high relative humidities are smaller than at low relative humidity, resulting from the reaction rate improvement. As in their investigation, the simulation values at a relative humidity of 100% in this work were smaller than those at a relative humidity of 85%. This is because the pressure drop in the stack has a great influence on the relative humidity and cell output parameters. Water generated by a higher operating temperature of 70 °C and a high relative humidity of 100% promotes the passage of protons.

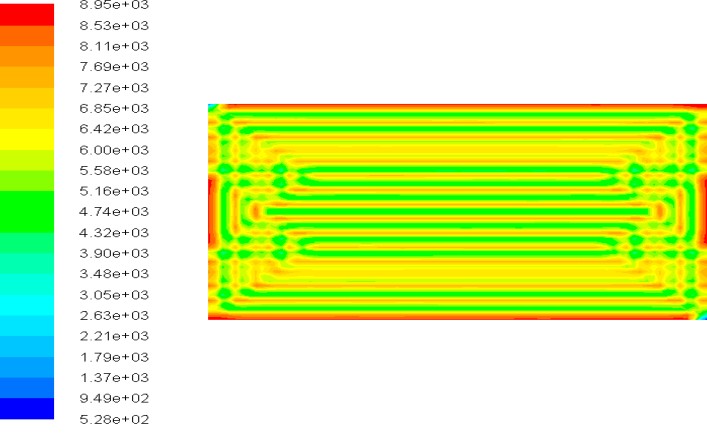

(**a**) Current density distribution at 85% relative humidity

**Figure 7.** *Cont.*

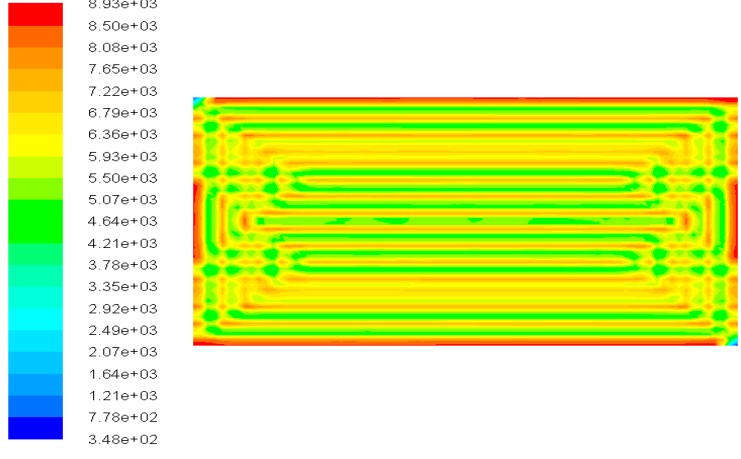

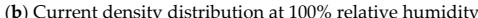

(**b**) Current density distribution at 100% relative humidity

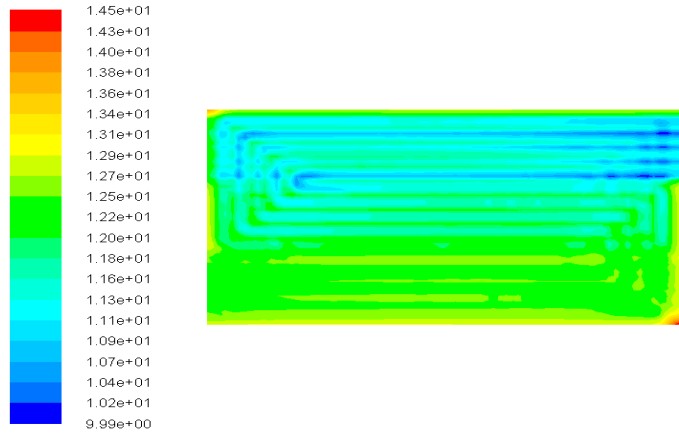

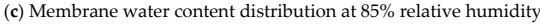

(**c**) Membrane water content distribution at 85% relative humidity

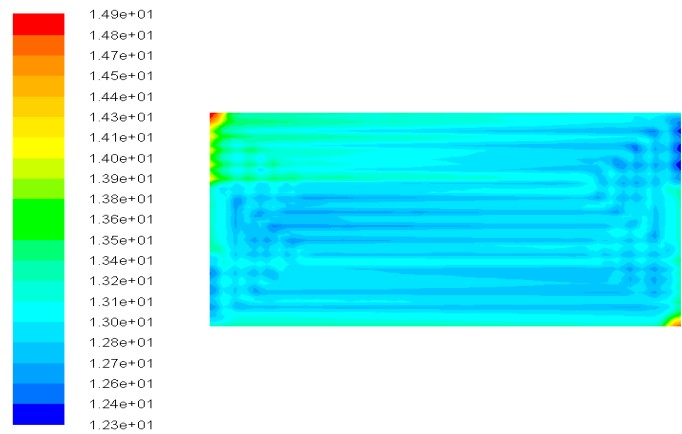

(**d**) Membrane water content distribution at 100% relative humidity

**Figure 7.** *Cont.*

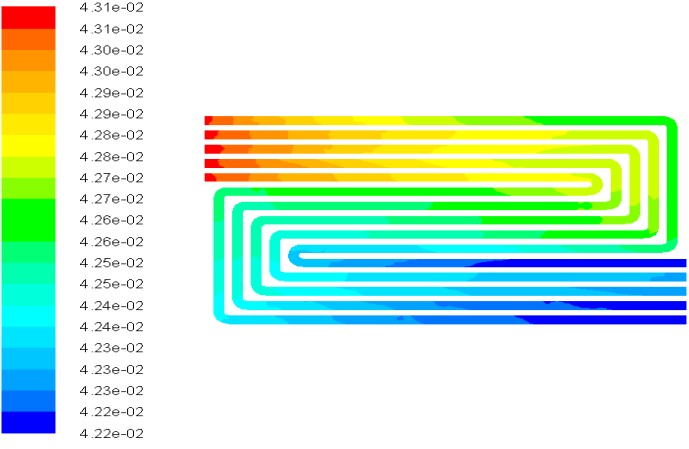

(**e**) Molar concentration of hydrogen at 85% relative humidity

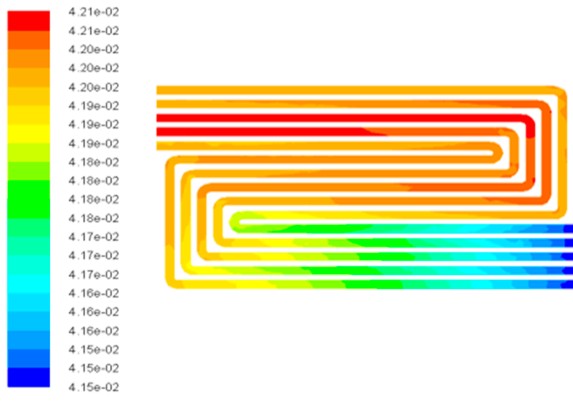

(**f**) Molar concentration of hydrogen at 100% relative humidity

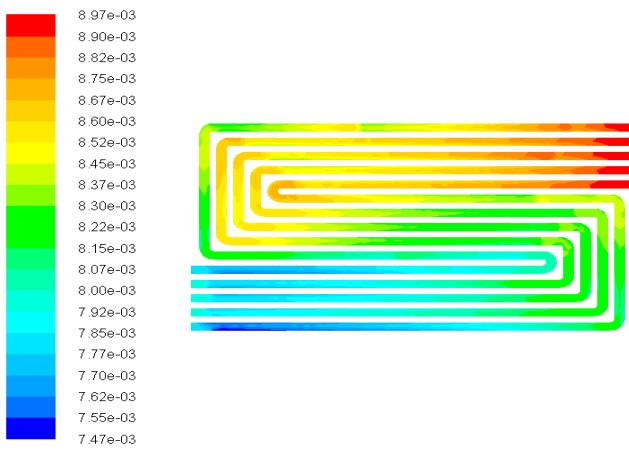

(**g**) Molar concentration of oxygen at 85% relative humidity

**Figure 7.** *Cont*.

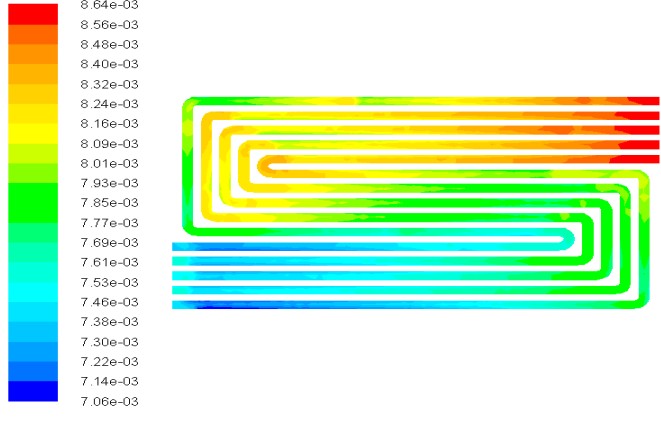

(**h**) Molar concentration of oxygen at 100% relative humidity

**Figure 7.** Contours of fuel cell species distribution.

## 5. Conclusions

In this study, the ARH model is applied to computational fluid dynamics to analyze the relative humidity effects on PEMFC performance. A three-dimensional model for the fastest power attenuation single cell with five serpentine channels is established in the stack. Experiments corresponding to simulation are performed. There are some conclusions, listed by polarization curves and counters of the fuel cell species distribution:

(1)  Single cell C10 power attenuation is the fastest, and its performance is the poorest under these experimental conditions.

(2)  The ARH model is valid because C10 experimental results and polarization curves predicted by the ARH model and the Fluent original model are consistent. ARH model results are closer to the experiment results, especially because its calculation deviation is almost 28% less than original model at a current density of 360 mA·cm$^{-2}$ for a relative humidity of 85%.

**Author Contributions:** Y.L. proposed the idea of visualization experiment combined with the simulation experiment. J.G. designed the experiments and performed the experiments. S.Y. analyzed the data. N.W. and J.G. accomplished the numerical simulation. Y.L. and J.G. wrote the paper.

**Funding:** This research was funded by the State Key Laboratory of Automotive Safety and Energy (KF1825), the Fundamental Research Funds for Beijing University of Civil Engineering and Architecture (X18083), the National Key Research and Development Program (2017YFB0102705 and 2016YFB0101305), National Natural Science Foundation of China (21676158), the Scientific Research Project of Beijing Educational Committee (KM201510016011), State Key Laboratory of Engines, Tianjin University (K2017-07), and the BUCEA Post Graduate Innovation Project (PG2018081). The APC was funded by BUCEA.

**Conflicts of Interest:** The authors declare no conflict of interest.

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
