# Peer review of "Asymptotic Analysis for the Effects of Anode Inlet Humidity on the Fastest Power Attenuation Single Cell in a Vehicle Fuel Cell Stack"

_applsci, doi:10.3390/app8112307_

Reviewer 1 Report

The manuscript entitled “Asymptotic Analysis for the Effects of Anode Inlet Humidity on the Fastest Power Attenuation Single Cell in a Vehicle Fuel Cell Stack” by Liu J. et al.

focuses on the three-dimensional and isothermal anode relative humidity (ARH) model, which is presented and used to study the anode inlet humidity effects on the fastest power attenuation single cell in a vehicle fuel cell stack. The pressure drop is considered into ARH model and saturation pressure is established by pressure drop. Based on pressure drop and relative humidity, simulations and tests are conducted by the authors.

There is a good agreement between the theoretical polarization curves predicted by the ARH model and the experimental data.

The manuscript is suitable to be published in this journal; however, some minor revisions should be made by authors before publication. 

1. Introduction

 The reviewer does recommend the authors to include the following relevant articles in the research field of proton and anion exchange membrane fuel cell system for automotive applications to broad readership and to improve the reference part:

1.   International Journal of Hydrogen Energy, 2014, 39(24), 12934-12947, DOI: 10.1016/j.ijhydene.2014.05.135;

2.   International Journal of Hydrogen Energy, 2017, 42(46), 28034-28047, DOI: 10.1016/j.ijhydene.2017.07.239;

3.   Fuel Cells, 2016, 16(5), 628-639, DOI: 10.1002/fuce.201500174;

4.   World Electric Vehicle Journal, 2016, 8(1), 131-138; DOI:10.3390/wevj8010131;

5.   Cogent Engineering, 2017, 4: 1357891, DOI: 10.1080/23311916.2017.1357891;

6.   International Journal of Hydrogen Energy, 2017, 42(33), 21158-21166, DOI: 10.1016/j.ijhydene.2017.06.209;

7. International Journal of Hydrogen Energy, 2017, 42(30), 19556-19575, DOI: 10.1016/j.ijhydene.2017.06.106.

2 Calculation

2.1. ARH model description  

At line 134 What is the Kano cycle? Can the authors insert a reference about this cycle?

2.3 Geometry and Mesh Generation   

At line 193 the authors should better motivate the hypothesis of incompressibility.

3.2 Experimental conditions

The sentence: “The ambient temperature of this experiment is 25and ambient air relative humidity is 30%.” is repeated twice at lines 232-233 and at lines 240-241.

4.3. Contours of fuel cell species distribution

At lines 397-399 the authors should declare the operative fuel cell voltage.

At lines 413-415 the sentences: “The hydrogen molar concentration is in the range of 4.22e-2 to 4.31e-2, and it gradually increases from anode channels inlet to outlet at relative humidity 85%. The similar tendency is that the hydrogen molar concentration gradually increases from 4.15e-02~4.21e-02 from anode channels inlet to outlet at relative humidity 100%.” appears not clear and it has to be better explained.

Author Response

Dear reviewer,

         We have revised the article in accordance with the comments.Thank you for your valuable advice.

Reviewer 2 Report

The authors report a 3D isothermal ARH model to study the effect of anode inlet humidity on the fastest power attenuation single cell in a vehicle fuel cell stack. The authors compared their ARH model to a Fluent model and found out that the former matches the experimental results better than the latter. This is a well written research article with interesting results. I recommend it for publication after addressing the following concerns

The authors need to add more details to their figure captions. In addition to including the information in the text, pointing the reader to what to look for in the figures is helpful.

The authors used serpentine channels in this study. How does the ARH model perform when flow channels with a different configuration (flow through type for example) were used? including a small paragraph discussing this would add more value to the manuscript.

Author Response

Dear reviewer,

         We have revised the article in accordance with the comments. Thank you for your valuable advice.
